# Paying Comprehensive Attention to the Temperature-Dependent Dual-Channel Excited-State Intramolecular Proton Transfer Mechanism of Fluorescence Ratio Probe BZ-DAM

**DOI:** 10.3390/ijms241813899

**Published:** 2023-09-09

**Authors:** Jiaan Gao, Yifu Zhang, Hongyan Mu, Min Yang, Xiaotong Guan, Guangyong Jin, Hui Li

**Affiliations:** Jilin Key Laboratory of Solid-State Laser Technology and Application, School of Physics, Changchun University of Science and Technology, Changchun 130022, China; gja13622147194@163.com (J.G.); if010522@163.com (Y.Z.); xuemuafm@163.com (H.M.); 18404361217@163.com (M.Y.); mianhuatang7657@163.com (X.G.)

**Keywords:** excited-state intramolecular proton transfer, density functional theory, potential energy curve, excited-state hydrogen bond dynamics, two different channels

## Abstract

The mechanism of fluorescence detection of diethyl chlorophosphate (DCP) based on 2-substituted benzothiazole (BZ-DAM) was studied by a theoretical calculation method. It should not be ignored that both the BZ-DAM and the detection product BZ-CHO have two excited-state intramolecular proton transfer (ESIPT) channels. Density functional theory (DFT) and time-dependent DFT (TDDFT) theory were used to study the photophysical mechanism of two compounds in two channels in (acetonitrile) ACN solvent, and the temperature dependence of the two channels was given. Channel 1 is more likely to exist at low temperatures and channel 2 is more likely to exist at high temperatures. By theoretical analysis of the constructed potential energy curve, the hydrogen bond energy and electron-hole analysis, we confirmed that both molecules undergo ESIPT and intramolecular charge transfer (ICT) processes in channel 1 and ESIPT and twisted intramolecular charge transfer (TICT) coupling processes in channel 2. The formation of product BZ-CHO molecules led to a significant fluorescence blue-shift phenomenon and inhibited the ICT process, which confirmed that BZ-DAM could be used as a fluorescence probe for fluorescence detection. We sincerely hope that this work will not only help to clarify the excited-state dynamics behavior of the BZ-DAM probe but also provide a new idea for designing and optimizing a new chemical dosimeter.

## 1. Introduction

Fluorescent probes find widespread application in detection and labeling techniques, including the determination of biomolecular content, metal ions, pesticide residues as well as the tracing of molecules [1,2,3,4,5,6,7,8,9,10]. Fluorescence probes based on excited-state intramolecular proton transfer (ESIPT) have been developed and designed [11,12,13,14,15]. ESIPT is an ultrafast process with dual fluorescence and large stokes shift properties [16,17,18,19,20]. Fluorescent probe molecules with ESIPT characteristics can be categorized into three types. We have illustrated these three classes of probes with examples and listed them in Appendix A. The first type is the “turn-off” fluorescent probe. The fluorescent probe molecules undergo ESIPT prior to the reaction, while the resulting product molecules do not exhibit ESIPT properties. The PBAS, L, 2, oPSDAN and HL^m^ were typical “turn-off” fluorescent probes. This type of fluorescent probe is employed for the sensitive and selective detection of various analytes. For instance, for the quantitative and qualitative detection of Cu^2+^, probe L was synthesized by Zhang et al. [21]. They demonstrated that the presence of Cu^2+^ significantly reduced the fluorescence intensity. A color fluorescence sensor oPSDAN was studied by Mehta et al. [22]. They verified that the sensor showed fluorescence quenching for Al^3+^. The second type is the “turn-on” fluorescent probe. The fluorescent probe molecules can be described more intuitively as follows: only the product molecules undergo ESIPT after the reaction occurs. The CPI, MZC-AC, BT-ITC, HBT-BChE, SNCN-AE and SNC-AE were typical “turn-on” fluorescent probes. This type of fluorescent probe is useful for monitoring reaction progress or identifying specific reaction products in complex mixtures. For example, the sensing mechanism of the probe BT-ITC was investigated by Kim et al. [23]. They confirmed the advantage of the probe in detecting H_2_S by its fluorescence “turn-on” response through ESIPT. The fluorescent probe HBT-BChE for bio-detection and imaging was developed by Pei et al. [24]. They found that the hydrolysis of HBT-BChE triggered a strong orange fluorescence signal through ESIPT, enabling fluorescence “turn-on”.

In particular, there is a class of probes in which both reactant and product molecules undergo the ESIPT process. As shown in Appendix A, such probe molecules generally have large stokes shifts. The Naph-1, HBT-Hg, BTMSP and BIN were typical ratio fluorescent probes. For instance, the BTMSP has been employed for detecting HCLO by Shen et al. [25]. They showed a fast response to HCLO, with a large stokes shift of 130 nm. Recently, Manas Kumar Das et al. synthesized benzothiazole-based fluorescence chemical dosimeter (BZ-DAM) molecules by the ESIPT process, and they can be successfully used to detect sarin in living cells [26]. Sarin is widely believed to be a nerve agent often used in terrorist attacks. Diethyl chlorophosphate (DCP) was used as a model compound for sarin due to its similar reactivity and lack of high toxicity. Based on the investigation of absorption and fluorescence spectra, they found that BZ-DAM showed a fluorescence-enhanced response to DCP at 540 nm in ACN solvent. It should not be ignored that there are double receptors (N1 and N2) in BZ-DAM and double receptors (N1 and O2) in BZ-CHO, which directly lead to the formation of two different conformations. Therefore, a thorough theoretical investigation is urgently needed to offer more insights into the ESIPT mechanism of BZ-DAM and BZ-CHO.

In this study, BZ-DAM was selected as the reactant and BZ-CHO as the product after adding DCP. Combined with the theoretical simulation results of DFT/TDDFT, the photophysical properties of BZ-DAM and BZ-CHO in (acetonitrile) ACN solvent under different electronic states in two channels were investigated. We obtained absorption spectra and fluorescence spectra, which provided the contribution of frontier molecular orbitals and major orbital transitions to electronic states. We formally described the ESIPT mechanism of DCP detection in two channels by Figure 1 and confirmed the occurrence of ESIPT by analyzing the potential energy curves of the S_0_ state and S_1_ state of BZ-DAM and BZ-CHO in ACN solvent and the time scale of the S_1_ state. Finally, two ESIPT processes at different temperatures are elucidated.

## 2. Results and Discussion

### 2.1. Optimized Geometric Structure and Infrared (IR) Vibration Spectra

The optimized structures of the different channels of BZ-DAM and BZ-CHO molecules in the different electronic states in (acetonitrile) ACN solvent are shown in Figure 1 and Figure 2. The corresponding changes in bond lengths, bond angles, and dihedral angles are listed in Table 1. The optimized percentage of BZ-DAM and BZ-CHO molecules at different temperatures is listed in Table 2 [27]. The presence of a double receptor (N1, N2) in BZ-DAM and the presence of a double receptor (N1, O2) in BZ-CHO led to the formation of two different conformers, as shown in Figure 1. As H1 approaches N1 of the two molecules, we define it as channel 1. When H2 tends to move to N2 and O2, we define it as channel 2.

Interestingly, when the temperature is low, the proportion of channel 1 in the two molecules (98.09% and 88.38%, respectively) is larger than that of channel 2 (1.91% and 11.62%, respectively). Both compounds are more likely to be present in channel 1 at low temperatures. In the first channel, the BZ-DAM molecule, the bond length of O1-H1 is elongated from 1.001 Å in the S_0_ state to 1.030 Å in the S_1_ state, while that of N1…H1 is shrunken from 1.641 Å (S_0_) to 1.536 Å (S_1_) with an enlargement of the bond angle σ (O1-H1…N1) from 150.06° to 151.75° after the photo-absorption process. The results confirm that the IHB (O1-H1…N1) is strengthened in the S_1_ state, which is conducive to the occurrence of the ESIPT pathway. Similarly, we have obtained the S_1_′ structure with an energy lower than that of the S_1_ structure and formed a new IHB (N1-H1…O1) whose length is 1.781 Å. For the BZ-CHO molecule, the bond length of N1…H1 is shrunken from 1.650 Å (S_0_) to 1.460 Å (S_1_) with an enlargement of the bond angle σ (O1-H1…N1) from 149.67° to 153.14°. The results confirm that the IHB is strengthened in the S_1_. In the S_1_′ structure, a new IHB (N1-H1…O1) was formed whose length is 1.822 Å, and the energy is lower than that of the S_1_ structure. These results indicate that the BZ-DAM and BZ-CHO molecules in the first channel probably undergo an effective ESIPT process in the ACN solvent. The stronger IHB of BZ-DAM in the S_1_′ structure compared to BZ-CHO proved that the BZ-CHO molecule is more stable after the addition of intracellular diethyl chlorophosphate (DCP).

However, at high temperatures (above 700 K), the proportion of channel 2 in the two molecules was larger (99.30% and 88.95%) than that of channel 1 (0.70% and 11.05%, respectively). It is shown that these two compounds are more likely to occur in the conformation of channel 2. Unlike channel 1, only the keto* structure exists in channel 2 for both molecules. In the BZ-DAM molecule, the IHB (O1-H2…N2) length of 1.658 Å in the S_0_ state. After photo-excitation, a new IHB (N2-H2…O1) is formed with a length of 1.670 Å in the S_1_ state. And, for the BZ-CHO molecule, the length of IHB (O1-H2…O2) in the S_0_ state is 1.658 Å. After photo-excitation, a new IHB (O2-H2…O1) is formed, which is 1.509 Å in the S_1_ state. The most noteworthy point is that the dihedral angle σ(C1-C2-C3-N1) of BZ-DAM changes by 43.3° (from −34.78° in the S_0_ state to 8.48° in the S_1_ state). And the dihedral angle σ(C1-C2-C3-N1) of BZ-CHO changes from 35.83° to −3.68° (a change of 39.5°). Therefore, we determined that the BZ-DAM and BZ-CHO in channel 2 probably undergo a coupling process of ESIPT and TICT in ACN. However, the geometric structures corresponding to the BZ-DAM and BZ-CHO in channel 1 both have good planarity. On the whole, the occurrence of TICT may lead to competition between radiative and non-radiative transition processes.

We simulated the IR spectra of the BZ-DAM and BZ-CHO compounds in the two channels, as shown in Figure 3. In the first channel, the vibration peaks of the BZ-DAM and BZ-CHO were changed from 3078 cm^−1^ and 3091 cm^−1^ (in the S_0_ state) to 2533 cm^−1^ and 2013 cm^−1^ (in the S_1_ state), and they were red-shifted 545 cm^−1^ and 1078 cm^−1^, respectively. This red-shift phenomenon indicates that the excited-state IHB is enhanced under light excitation. It is noteworthy that with the breaking of the old bond (O1-H1) and the formation of the new bond (N1-H1), it is confirmed that the ESIPT process of BZ-DAM and BZ-CHO occurred in the excited state. Analysis showed that IHB was enhanced after photo-excitation, which was conducive to the occurrence of the ESIPT reaction. The red-shift of BZ-CHO is greater than that of BZ-DAM, indicating that the addition of DCP can more significantly enhance the intensity of IHB in the excited state. In the second channel, the breaking of the old bonds (N2-H2 for BZ-DAM and O2-H2 for BZ-CHO) and the formation of the new bond (BZ-DAM and BZ-CHO: O1-H2) confirmed that the ESIPT process occurs in BZ-DAM and BZ-CHO.

### 2.2. Absorption and Fluorescence Spectra

In order to understand the optical and physical properties of BZ-DAM after adding DCP, the absorption and fluorescence spectra of BZ-DAM and BZ-CHO in the two channels are simulated by using the MPW1PW91/6-31G (d, p) theoretical level as shown in Figure 4.

The calculated maximum absorption peaks of BZ-DAM in the two channels are located at 397 nm and 400 nm, respectively, which is in good agreement with the experimental value (400 nm). Thus, we determined that there could indeed be two distinct pathways to induce the ESIPT process. The maximum absorption peak of BZ-CHO in both channels was 346 nm. Compared with BZ-DAM, the absorption peak of BZ-CHO showed a blue-shift of 50 nm. In channel 1, in the S_1_ state of BZ-DAM and BZ-CHO correspond to short emission peaks of 455 nm and 406 nm, respectively. And the long fluorescence peaks of BZ-DAM and BZ-CHO correspond to the proton transfer S_1_′ states at 586 nm and 495 nm, respectively. The stokes shifts of the BZ-DAM and BZ-CHO molecules are 189 nm and 149 nm. It is noteworthy that BZ-DAM and BZ-CHO have only one fluorescence peak in channel 2, which is attributed to the proton transfer states. The fluorescence peaks of BZ-DAM and BZ-CHO are 626 nm and 545 nm, respectively, accompanied by stokes shifts of 226 nm and 199 nm. Therefore, the BZ-DAM molecule of the two channels has larger stokes shifts. The fluorescence peaks of the two compounds observed in the experiment are 595 nm and 540 nm, respectively. Compared with BZ-DAM, the fluorescence peak of BZ-CHO showed a blue-shift of 91 nm and 81 nm due to the occurrence of the ESIPT process in the two channels, which confirmed that BZ-DAM could be used as a fluorescence probe for fluorescence detection. Compared with channel 1, the fluorescence intensity decreased due to the TICT process in channel 2.

In addition, in order to study the fluorescence properties of the two compounds in the two channels, the fluorescence lifetime and fluorescence rate of the two forms of the S_1_ state and the S_1_′ state were calculated according to the formulas in the literature [28], which are listed in Table 3. Similarly, the longer the fluorescence lifetime, the lower the fluorescence rate. It is obvious that BZ-DAM-S_1_′ in channel 1 and BZ-DAM-S_1_ in channel 2 have longer fluorescence lifetimes and lower fluorescence rates. The BZ-DAM molecule has good optical properties because of the large stokes shift.

### 2.3. FMOs and Hole-Electron Analysis

It is worth commenting on the appearance of double absorption peaks: the maximum absorption peak is almost entirely from the first singlet transition (S_0_ → S_1_), corresponding to the highest occupied molecular orbitals (HOMOs) to the lowest unoccupied molecular orbitals (LUMOs), as expected, and the lower intensity absorption peak is mainly from the S_0_ → S_4_ transition (see Appendix A). The transition percentages (S_0_ → S_1_) are 64.20%, 69.33%, 66.28% and 69.37% in BZ-DAM, BZ-CHO, BZ-DAM-1 and BZ-CHO-1, respectively. In order to analyze the redistribution trend of electrons in the structure, we divided BZ-DAM and BZ-CHO into fragment 1 and fragment 2. As shown in Figure 5, the Hirshfeld method was used to calculate the electron density. The relevant electron densities are listed in Table 4.

In channel 1, for fragment 1, the electron densities of the two compounds decrease from 95.88% and 65.87% of HOMO to 92.63% and 59.96% of LUMO, respectively. In fragment 2, the electron density increases from 4.12% and 34.13% of HOMO to 7.37% and 40.04% of LUMO. In channel 2, the electron densities of the two compounds increase from 92.85% and 60.70% of HOMO to 97.81% and 75.08% of LUMO, respectively, in fragment 1. In fragment 2, the electron density decrease from 7.15% and 39.30% of HOMO to 2.19% and 24.92% of LUMO. At the same time, since the HOMO difference between Fragment 1 and Fragment 2 of BZ-DAM is the largest, we believe that the BZ-DAM molecules have more obvious ICT characteristics, which may be more conducive to the occurrence of the ESIPT process.

In addition, in channel 1, the electron density of the O1 atom decreases from 6.40% to 0.54% and that of the N1 atom increases from 0.37% to 1.39% after photo-excitation in BZ-DAM. The electron density of the O1 atom decreases from 11.79% to 0.22% and that of the N2 atom increases from 4.47% to 8.52% after photo-excitation of BZ-CHO. Likewise, the two molecules also have the same change trend in channel 2. In the excited states, the electron density of the donor atom decreases while that of the acceptor atom increases, which promotes the ESIPT process of the H1 atom transfer from the O1 atom to the N1 atom and provides the driving force for proton transfer.

### 2.4. Reduced Density Gradient (RDG) and HB Strength

In order to verify the variation of the IHBs intensity during proton transfer, RDG is also calculated [29,30], as shown in Figure 6, Figure 7 and Appendix A. HB interactions, van der Waals interactions and the steric crowding effect correspond to blue, green and red colors, respectively. In channel 1, the spikes of BZ-DAM and BZ-CHO are located at −0.060 a. u. and −0.055 a. u. in the S_0_ state, which shifts to the more negative −0.085 a. u. and −0.055 a. u. region in the S_1_ state. The spike peak is assigned to N1…H1-O1. This means that the IHBs of BZ-DAM and BZ-CHO are enhanced under light excitation, which will promote the ESIPT reaction. In the S_1_′ state, the spikes of BZ-DAM and BZ-CHO are located at −0.043 and −0.040 a. u., respectively. The spike peak is assigned to O1…H1-N1. In channel 2, the spikes of BZ-DAM and BZ-CHO are located at −0.055 and −0.055 a. u. in the S_0_ state. The spike peak is assigned to N2…H2-O1 and O2…H2-O1, respectively. However, the spikes of BZ-DAM and BZ-CHO are located at −0.060 and −0.080 a. u. in the S_1_ state. The spike peak is assigned to O1-H2…N2 and O1-H2…O2, respectively. The formation of new IHB confirms that the ESIPT process occurs in both compounds under both channels.

In order to further clarify the difference in the IHBs interaction between BZ-DAM and BZ-CHO in ACN solvent, we also predicted the value of the hydrogen bond energy (E_HB_) by simulating the bond critical point (BCP) [31], which is listed in Table 5. Based on previous similar conclusions, we reasoned that the enhanced behavior of HB induced by photo-excitation largely reflects the developing trend of the ESIPT response. Generally speaking, the larger the |E_HB_| value, the stronger the corresponding HB. As can be seen from Table 6, in channel 1, the S_0_ state |E_HB_| of the two compounds is in the range from 2.5 kcal/mol to 14.0 kcal/mol; at this time, the electrostatic action plays a dominant role. The S_1_ state |E_HB_| of the two compounds is greater than 15.0 kcal/mol, indicating that the main characteristics of the HB interaction are electrostatic and induction, and dispersion only has a supplementary role. The results showed that the IHB force of both compounds in channel 1 was enhanced under the action of photo-excitation. In channel 2, for BZ-DAM-1, the |E_HB_| in the S_0_ state is less than 2.5 kcal/mol when electrostatic and dispersion are dominant. The |E_HB_| of BZ-DAM-1 in the S_1_ state ranges from 2.5 to 14.0 kcal/mol. BZ-CHO-1 shows the exact opposite trend. The above analysis of E_HB_ is consistent with that of RDG.

The ICT properties corresponding to the S_1_ → S_0_ transition were further quantitatively calculated by electron-hole analysis and charge density difference (CDD) (see Figure 8) [32,33]. The relevant data are listed in Appendix A. The red arrows correspond to the distance from the center of mass of the electron to the hole (denoted by the D value). t-index > 0 implies that charge transfer (CT) excitation results in a sufficient hole and electron separation. Combined with Appendix A, it is found that the D index of BZ-DAM-keto in channel 1 is the largest, which is 1.507 Å. In channel 2, the D index of BZ-DAM-1-keto is the largest, which is 3.960 Å and t-index > 0. The separation of BZ-DAM-1-keto was confirmed to be the most sufficient. These data indicate that BZ-DAM has the stronger ICT effect.

### 2.5. Excited-State Proton Transfer Mechanism and Dynamic Simulation

In order to further study the ESIPT process of the BZ-DAM and BZ-CHO compounds in the two channels, we also studied the molecular potential energy curve (PECs), as shown in Figure 9. The S_0_ state was accompanied by an impassable barrier and a high barrier of 7.366 kcal/mol. However, in the S_1_ state, there was no barrier or a 0.710 kcal/mol low barrier between the two compounds in the two channels. It proves that the ESIPT process is more likely to occur in the excited state than in the ground state. Similarly, the high barrier of the reverse proton transfer (RPT) process between the two compounds in both channels confirms that RPT cannot occur. Therefore, we have verified that the ESIPT process is irreversible. This property of the BZ-DAM molecule has great advantages for DCP detection.

In addition, we investigated the time scales of the two compounds in both conformations using Bonn–Oppenheimer molecular dynamics (BOMD), as shown in Figure 10. We give the time of proton transfer between the two molecules. In channel 1, the ESIPT process of the BZ-DAM molecules occurred at 131.7 fs, while that of the BZ-CHO molecules occurred at 152.7 fs. In channel 2, in BZ-DAM and BZ-CHO, the ESIPT process is experienced at 215.2 fs and 33.5 fs, respectively. In contrast, the BZ-CHO molecule of channel 2 has the fastest ESIPT process.

The detection mechanism at different temperatures can be illustrated in Figure 11, which will help to synthesize more efficient fluorescent probes in the future. Channel 1 is more likely to undergo an ESIPT + ICT process at low temperatures. Channel 2 is more likely to undergo an ESIPT + TICT coupling process at high temperatures. The formation of the product BZ-CHO molecule led to a significant fluorescence blue-shift phenomenon, which confirmed that BZ-DAM could be used as a fluorescence probe for detection.

## 3. Materials and Methods

All calculations are performed using Gaussian programs in this work [34]. We calculated the absorption peak of BZ-DAM in ACN solvent using six different functionals and the 6-31G (d, p) basis set in Table 5. The absorption peaks of BZ-DAM obtained by the MPW1PW91 functional are separately located at 310 nm and 397 nm, which coincided well with the experimental data (311 nm and 400 nm). The accuracy of theoretical data is confirmed. Therefore, we use the MPW1PW91/6-31G (d, p) [35]. The configuration optimization of BZ-DAM and BZ-CHO molecules in different electronic states by the DFT [36,37,38] and TD-DFT [28,39,40,41,42] methods in the two channels was carried out without any constraints. All the vibration analysis results have no virtual frequency, which proves that all the optimized structures are local minima. Moreover, we used the solvent effect of ACN and considered utilizing the polarizable continuum model (PCM) by applying the integral equation formalism variant (IEFPCM). For the study of excited-state hydrogen bonding dynamics in two compounds, the classical trajectory calculation (Bonn-Oppenheimer molecular dynamics (BOMD)) is implemented in the S_1_ state [43,44]. In addition, Multiwfn software [45] is used to analyze the frontier molecular orbitals (FMOs), reduced density gradient (RDG) [46] and electron-hole analysis. VMD software is used for visualization [47].

## 4. Conclusions

In conclusion, this work investigates the photo-physical mechanism of two compounds in ACN solvent by DFT/TDDFT theory and gives the likelihood of temperature dependence in the two channels. Channel 1 is more likely to exist at low temperatures (less than 700 K), and channel 2 is more likely to exist at high temperatures (above 700 K). By theoretical analysis of the constructed PECs, E_HB_ and electron-hole analysis, we confirm that an ESIPT + ICT process occurs in channel 1, and an ESIPT + TICT coupling process occurs in channel 2. Moreover, we give the time scale of proton transfer between the two molecules. In channel 1, the ESIPT process occurred at 131.7 fs and 152.7 fs. In channel 2, the ESIPT process occurred at 215.2 fs and 33.5 fs in BZ-DAM and BZ-CHO, respectively. Furthermore, the formation of the BZ-CHO molecule led to a significant fluorescence blue-shift phenomenon and inhibited the ICT process, which confirmed that BZ-DAM could be used as a fluorescence chemical dosimeter probe for detecting DCP. This work elucidates the ESIPT mechanism of BZ-DAM at different temperatures. The results show that BZ-DAM is a reliable and useful tool for the selective detection of DCP. Our work can effectively contribute to the optimization and design of more efficient chemical dosimeters with ESIPT characteristics, thereby prompting the development of fluorescent materials for diverse applications in the future.

## Data Availability

Not applicable.

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
