# Peer review of "Paying Comprehensive Attention to the Temperature-Dependent Dual-Channel Excited-State Intramolecular Proton Transfer Mechanism of Fluorescence Ratio Probe BZ-DAM"

_ijms, 2023, doi:10.3390/ijms241813899_

Round 1

Reviewer 1 Report

Dear Authors,

The subject of this manuscript was a synthesis of a new fluorescence chemical dosimeter (BZ-DAM) with an intramolecular hydrogen bond.  The paper uses a comprehensive approach to the subject matter and applies the theoretical investigation to offer more insights into the ESIPT mechanism of BZ-DAM. In this work, the authors used density functional theory (DFT) and time-dependent DFT (TDDFT) theory to investigate two compounds' photo-physical mechanism in ACN solvent and give the temperature-dependent in the two channels. Channel 1 was more likely to exist at low temperatures, and channel 2 was more likely to exist at high temperatures. By theoretical analysis of the constructed potential energy curve, hydrogen bond energy and electron-hole analysis were confirmed that both molecules undergo ESIPT and intramolecular charge transfer (ICT) process in channel 1 and ESIPT and twisted intramolecular charge transfer (TICT) coupling process in channel 2.

The paper is written in exciting language, but a certain dissatisfaction is left by the summary and perspectives related to using the developed research methodology in designing more efficient and accurate sensors in the future. I would ask you to add such a summary, taking into account the potential applications of such consciously designed molecules.

Accordingly, I request that the manuscript be accepted for publication after minor revisions.

Dear Authors,

The subject of this manuscript was a synthesis of a new fluorescence chemical dosimeter (BZ-DAM) with an intramolecular hydrogen bond.  The paper uses a comprehensive approach to the subject matter and applies the theoretical investigation to offer more insights into the ESIPT mechanism of BZ-DAM. In this work, the authors used density functional theory (DFT) and time-dependent DFT (TDDFT) theory to investigate two compounds' photo-physical mechanism in ACN solvent and give the temperature-dependent in the two channels. Channel 1 was more likely to exist at low temperatures, and channel 2 was more likely to exist at high temperatures. By theoretical analysis of the constructed potential energy curve, hydrogen bond energy and electron-hole analysis were confirmed that both molecules undergo ESIPT and intramolecular charge transfer (ICT) process in channel 1 and ESIPT and twisted intramolecular charge transfer (TICT) coupling process in channel 2.

The paper is written in exciting language, but a certain dissatisfaction is left by the summary and perspectives related to using the developed research methodology in designing more efficient and accurate sensors in the future. I would ask you to add such a summary, taking into account the potential applications of such consciously designed molecules.

Accordingly, I request that the manuscript be accepted for publication after minor revisions.

Reviewer 2 Report

The paper G.Jin and H. Li deals with huge theoretical studying fluorescence mechanism (ESIPT especially). Theoretical calculation and mechanism investigation (IR etc) was carried out on a high level as well as its discussion.

There are a lot of significance meaning issues (double meaning, general sentente, etc). I suggest to strongly rework and resubmit the paper.

Title is too general. I suggest to specify it.

Abstract.  Authors claimed “A new fluorescence chemical dosimeter (BZ-DAM) with intramolecular hydrogen bond has been synthesized for testing the nerve agent sarin, and the tested product is BZ-CHO molecule.” The BZ-DAM and BZ-CHO are NOT NEW, they have been synthesized by other group https://doi.org/10.1039/D2NJ04260F , this is not the extension of previous group work. This is a huge misunderstanding and it make readers to be confused.

Abstract. Authors claimed “…with intramolecular hydrogen bond has been synthesized for testing the nerve agent sarin….” But there is no sarin testing in the paper. Authors should explain in detail what they mean or remove the uncertain meaning.

Introduction. Authors did a good literature searching and review e.g.“….Kumar et al. developed a luminescent sensing probe Tb-SA based on doped erbium optical fibers in a mixed solvent of protonic and non-protonic water…” and “….Tohora et al. designed a ESIPT-based fluorescent "off-on-off" probe (NT) for detection of in H2O/DMSO medium….” and “….Kim investigated the sensing mechanism of the BT-ITC probe for detecting H2S…” and “…Pei et al. developed the fluorescent probe HBT-BChE for bio-detection and imaging….” etc. I suggest to illustrate the mentioned compounds in introduction.

Why do Authors write in abstract about sarin detection, but they choose diethyl chlorophosphate for detection?

All Schemes should be reworked. All subscript should be transformed to superscript, for example Scheme 1 BZ-DAM Authors indicate (left row, first) the N2 O1 N1 as second nitrogen, first oxygen and first nitrogen, respectively, but for me and I think for most of the readers the N2 means two nitrogen N=N or similar. All of these should be N2 O1 N1  à N2 O1 N1  Same issue has been detected on other Schemes and Figures.

Authors claimed in title “detection mechanism for ESIPT-ICT-based”, as I understood correctly the BZ-DAM reacts with DCP and BZ-CHO was obtained (Line 66 confirm that), but there is a covalent detection, whereas ESIPT affect to fluorescence. In this way the studying of ESIPT effect in BZ-DAM has no meaning.

Addition to the previously issues. The detection scheme should be provided.

Round 2

Reviewer 2 Report

I am pleased with how Authors rewrite the title and added figure 11 that summarized the work. I have no issues according to experiments. When I am looking from the side of readers, I see several confusing points. Unfortunately, most of my comments from previous revision have not been taken into account. I suggest to accept the article after carefully revising the following issues:

1.      Abstract is still not correlated with manuscript. Abstract should summarize what have been done in this work. Remove the sentence about new synthesized compound. I recommend to start abstract like this: “The mechanism of fluorescence detection of DCP by chemical dosimeter BZ-DAM & BZ-CHO based on 2-substituted benzathiazole has been comprehensively studied using theoretical calculation and experiments”. Please, feel free to rewrite it in your own style.

2.      “A new fluorescence chemical dosimeters (BZ-DAM) with intramolecular hydrogenhydrogen bond has been synthesized by Manas Kumar Das et al for testing the nerve agent sarin, and the tested product is BZ-CHO molecule”. In paper https://doi.org/10.1039/D2NJ04260F Authors studied BZ-DAM to detect DCP (undoubtedly, DCP is closely reactive to sarin) and CAN BE USED but not HAVE BEEN USED to detect sarin. According to your sentence the reader might understand that BZ-DAM was used to detect sarin, but there is no investigation about applying BZ-DAM in sarin detection. Authors have well written without double meaning this problem in Line 154-156, but the abstract is still confused.

3.      Point 4. I thanks to Authors that they find mistakes in previously discussion, but my main idea was neglected: “I suggest to illustrate the mentioned compounds in introduction”. I think that is very useful for readers.

4.      Point 5 and Response 5. Authors provided 6 articles which contain a description of excited states with subscript. I have completely evaluated suggested articles:

a.       https://doi.org/10.1016/j.dyepig.2022.110192 No subscript has been found, the O1 N1 spelling was used

b.      Chinese Journal of Chemical Physics (2023) 1–10 No article with page 1-10 has been found https://pubs.aip.org/cps/cjcp/issue/36/1

c.      https://doi.org/10.1016/j.saa.2022.122141 Subscript has been found

d.      https://doi.org/10.1016/j.jphotochem.2022.113799 Subscript has been found

Due to a lack of examples of using subscript, I still consider that subscript might confuse the readers, I strongly suggest you to rework schemes according to https://doi.org/10.1016/j.dyepig.2022.110192 (O1 N1 etc) or accept my previous comment with superscript.

Here is one of the series of confusing examples of using subscript. If you can not see the picture here, please download a pdf-version of answers.

Grammar and syntax mistake:

Keywords. Remove the digits.

Figure 11. Fix “bule-shift”

Round 3

Reviewer 2 Report

I am thankful to Authors for accurately considering my comments. I suggest to accept the article in present form.